# Copy Number Variations and Expression Levels of Guanylate-Binding Protein 6 Gene Associated with Growth Traits of Chinese Cattle

**DOI:** 10.3390/ani10040566

**Published:** 2020-03-27

**Authors:** Dan Hao, Xiao Wang, Bo Thomsen, Haja N. Kadarmideen, Xiaogang Wang, Xianyong Lan, Yongzhen Huang, Xinglei Qi, Hong Chen

**Affiliations:** 1College of Animal Science and Technology, Northwest A&F University, Shaanxi Key Laboratory of Animal Genetics, Breeding and Reproduction, Yangling 712100, Shaanxi, China; haodan111121@163.com (D.H.); wangxiaogang0401@163.com (X.W.); lan342@126.com (X.L.); hyzsci@126.com (Y.H.); 2Department of Molecular Biology and Genetics, Aarhus University, 8000 Aarhus C, Denmark; bo.thomsen@mbg.au.dk; 3Department of Applied Mathematics and Computer Science, Technical University of Denmark, 2800 Kongens Lyngby, Denmark; xiwa@dtu.dk (X.W.); hajak@dtu.dk (H.N.K.); 4Bureau of Animal Husbandry of Biyang County, Biyang 463700, Henan, China; byq273@126.com

**Keywords:** cattle, *GBP6*, CNVs, growth traits

## Abstract

**Simple Summary:**

Our study identified the copy number variations (CNVs) and transcriptional expressions of guanylate-binding protein 6 (*GBP6*) in Chinese cattle, as well as analyzed the association of CNVs and expressions with the growth traits of Chinese cattle. The results showed Xianan cattle with gain types (Log_2_2^−ΔΔCt^ > 0.5) of CNVs of *GBP6* had highest relative gene expression levels in the muscle tissues and displayed superior phenotypic values of body weight, cannon circumference and chest circumference. Our study suggested that CNV gain types of *GBP6* could be used as the candidate markers in the cattle-breeding program for growth traits.

**Abstract:**

Association studies have indicated profound effects of copy number variations (CNVs) on various phenotypes in different species. In this study, we identified the CNV distributions and expression levels of guanylate-binding protein 6 (*GBP6*) associated with the growth traits of Chinese cattle. The results showed that the phenotypic values of body size and weight of Xianan (XN) cattle were higher than those of Nanyang (NY) cattle. The medium CNV types were mostly identified in the XN and NY breeds, but their CNV distributions were significantly different (adjusted *p* < 0.05). The association analysis revealed that the body weight, cannon circumference and chest circumference of XN cattle had significantly different values in different CNV types (*p* < 0.05), with CNV gain types (Log_2_2^−ΔΔCt^ > 0.5) displaying superior phenotypic values. We also found that transcription levels varied in different tissues (*p* < 0.001) and the CNV gain types showed the highest relative gene expression levels in the muscle tissue, consistent with the highest phenotypic values of body weight and cannon circumference among the three CNV types. Consequently, our results suggested that CNV gain types of *GBP6* could be used as the candidate markers in the cattle-breeding program for growth traits.

## 1. Introduction

Structural variations (SVs) in the genome refer to DNA sequence polymorphisms in the fragment lengths with a dozen or hundreds of bases, which have larger effects on phenotypic variations than SNPs [1]. As one category of SVs, copy number variations (CNVs) are defined as the type of large segments of DNA that are repeated and they vary in copy-number [2]. Generally, CNVs include insertions, deletions and duplications in a simple or complex structure, such as gains or losses of heterozygous or homologous sequences at multiple sites [3]. Single nucleotide polymorphisms (SNPs), the single nucleotide genetic variation in the genome, enable most of the common variations to be captured and genetic changes related to complex traits to be detected by many association studies [4]. However, a great difference between different individual genomes was caused by CNVs, which cover more base pairs than the per-locus mutations of SNPs and result in dramatic phenotypic consequence [5,6].

Recently, association studies indicate that CNVs have profound effects on various phenotypes including diseases (e.g., neurological and developmental diseases) [7,8] and non-pathogenic traits (e.g., height) [9] in humans. In domestic animal species, researchers have summarized the associations of CNV with pea-comb phenotype, late feathering, dark brown plumage and dermal hyperpigmentation in chickens [10]. Yang et al. [11] also found that several copy number variation regions (CNVRs) were involved in fetal muscle development, prostaglandin synthesis and bone color traits of sheep. Since the first CNV snapshot analysis identified 37 CNVRs in the pig genome across chromosomes 4, 7, 14, and 17 [12], a number of reports have identified CNVs related to carcass [13], disease [14], meat quality [15] and fatty acid composition [16] traits in pigs. Similarly, a series of CNV studies in cattle suggested a relationship with milk production [17], meat tenderness [18], immune system [19], feed efficiency [20] and fertility [21] traits. In beef cattle, 17 CNVs were identified to associate with seven different body traits [22]. Moreover, CNVs of ten genes (*CYP4A11*, *GBP2*, *GPC1*, *KCNJ12*, *KLF3*, *KLF6*, *LEPR*, *MAPK10*, *MICALL2* and *MYH3*) were identified in correlation with transcript expression levels and growth traits (e.g., body length and body weight) of Chinese domestic cattle in the previous CNV studies [23,24,25,26,27,28,29,30,31,32].

Guanylate-binding protein (GBP) with a molecular weight of 67–73 kilodalton (kDa) belongs to the superfamily of guanosine triphosphatases (GTPases) that could hydrolyze guanosine-5’-triphosphate (GTP) to guanosine-5’-diphosphate (GDP) and guanosine-5’-monophosphate (GMP) [33]. GBPs play the important roles in multitude of cellular processes (e.g., signal transduction, translation, vesicle trafficking, and exocytosis) [34] and have functions on the membrane of host and pathogen to mediate cell-autonomous host resistance in some infections [35]. Previous studies have revealed that the CNVs of the GBPs (e.g., *GBP2* and *GBP4*) were associated with growth traits in Chinese domestic cattle [24,36]. In addition, *GBP6* was found in an overlap with CNV204 which was significantly associated with growth trait of Nellore cattle at the weaning stage [22]. Furthermore, *GBP6* is located in CNVR37 that were overlapped with the quantitative trait loci (QTLs) of meat and carcass traits in Chinese bulls [37]. Transcriptome analysis also revealed that *GBP6* was differently expressed in two kinds (i.e., tough and tender) of longissimus dorsi in Angus cattle. The expression level of *GBP6* was up-regulated in the tough beef tenderness [38].

Nevertheless, few studies are conducted to interpret the relationships between CNVs and expression levels of *GBP6* associated with growth traits in Chinese domestic cattle. Therefore, the aim of this study was to identify the *GBP6* CNVs and expressions in six different Chinese beef cattle breeds, and to reveal their possible associations with growth traits, which could provide valuable molecular information regarding the usage of *GBP6* in the breeding programs of beef cattle.

## 2. Materials and Methods

### 2.1. Animals and Phenotypes

A total of 524 Chinese female beef cattle in six breeds were involved in this study, including 112 Nanyang (NY) cattle, 105 Qinchuan (QC) cattle, 213 Xianan (XN) cattle, 32 Denan (DN) cattle, 32 Xiajia (XJ) cattle and 30 Zaosheng (ZS) cattle. They were collected from six farms of six regions, respectively (Figure 1). The six farms were mainly situated in the central and northwest regions of China (Figure 1). The map of three provinces (i.e., Gansu, Henan and Shaanxi) was realized by R package *maptools* (version 0.9-5) using geographic information system (GIS) basic data of China. The average temperatures (°C) and precipitations (mm) were calculated using the data over past thirty years from the national meteorological information center of China (http://data.cma.cn).

Using the measurement methods of Gilbert et al. [39], we recorded twelve phenotypes for NY, QC and XN cattle that included body height (BoH, cm), body length (BoL, cm), body weight (BoW, kg), cannon circumference (CaC, cm), chest circumference (ChC, cm), chest depth (ChD, cm), chest width (ChW, cm), cross height (CrH, cm), hucklebone width (HuW, cm), rump length (RuL, cm) and waist circumference (WaC, cm). Only four phenotypes (i.e., BoH, BoL, BoW and ChC) were measured in the animals of NY, QC and XN breeds. However, no phenotypic measurement was performed for the other breeds (i.e., DN, XJ and ZS) due to the commercial limitations on these farms. All cattle were measured at 2 years of age, when they were ready for harvest in a commercial slaughterhouse.

### 2.2. DNA and RNA Extractions

All animal experiments were approved by the Animal Care Commission of the College of Veterinary Medicine, Northwest A&F University (Permit Number: NWAFAC1019). Blood samples were collected from each animal through the jugular vein into a vacuum tube. Additionally, we selected 12 cattle including 9 QC cattle in three age stages (i.e., fetal at 90 days of gestation, newborn calves and adult cattle; each stage included 3 cattle) with three replicates at each age stage and 3 XN cattle only in the adult age stage due to commercial limitations. Then, seven tissue samples (i.e., heart, kidney, liver, longissimus dorsi muscle, lung, spleen and subcutaneous fat) with three replicates were collected from each age stage of each selected animal from the QC and XN breeds. After the identification of CNV type, another 9 tissue samples of longissimus dorsi muscle were collected from XN adult cattle for the further gene expression validation of growth traits. Blood and tissue samples were immediately stored in the liquid nitrogen for genomic DNA and RNA extractions, respectively.

As TRIzol or TRI reagent is advantageous to RNA purification [40], this study used TRIzol (Takara, Dalian, China) to treat RNA as RNase-free DNase for RNA extraction from tissue samples. Then, 1% agarose gel electrophoresis and spectrophotometry measurement were conducted to estimate the density and the purity of RNA. With 1μg RNA as the template, StarScript II One-step RT-PCR Kit with genomic DNA Eraser (GenStar, Beijing, China) resulted in reverse transcription in complementary DNA (cDNA).

### 2.3. Primers Design for Quantitative Polymerase Chain Reaction Amplification

Two pairs of primers on *GBP6* for quantitative polymerase chain reaction (qPCR) amplification were designed by Primer Premier software (version 6.2.4) (http://www.premierbiosoft.com). The first and second primer pairs were located in the first intron and third exon regions of *GBP6*, respectively (Figure 2). The genomic location of *GBP6* (AC_000160.1) was identified based on *Bos taurus* genome of Bos_taurus_UMD_3.1 version. The genomic primer positions on the gene structure of *GBP6* were visualized by Illustrator for Biological Sequences (IBS) software (version 1.0). The correlation coefficient of the two primers of *GBP6* in the PCR efficiency was nearly 100%. In addition, the results of CNV identification using two primers were the same, thus, we chose the second primer in the third exon region for the CNV and expression results comparisons. The primers of *GBP6* and reference genes used for relative expression levels were listed in Table 1.

The amplification efficiencies of primers were tested by genomic DNA with PCR products concentration gradients that were 500, 100, 20, 4 and 0.8 ng. Afterward, 13 μL 2 × RealStar Green Power Mixture (GenStar, Beijing, China), 25 ng extracted DNA or RNA, 10 pmol of primers and 9 μL H_2_O consisted of 25 μL volume of the reaction mixture for the response system. The standard procedure for qPCR reaction was 95 °C for 10 min followed by 40 cycles of 95 °C for 15 s and 60 °C for 1 min. Additionally, the melting curve was added automatically by the CFX 96 TM Real-Time Detection System (Bio-Rad, Hercules, California, USA).

### 2.4. Identification of CNV and Expression Levels for GBP6

The relative CNV copy numbers were defined by an evaluation of 2 × 2^−ΔΔCt^ with the average threshold cycles (Ct) in triplicate independent repeats, where $\Delta \mathrm{Ct}={\mathrm{Ct}}_{\mathrm{target}\;\mathrm{gene}}-{\;\mathrm{Ct}}_{\mathrm{reference}\;\mathrm{gene}\;}\;$ following the previous studies [26,30,31,41,42,43,44,45,46]. The basic transcription factor 3 (*BTF3*) acted as the internal reference gene following the study of Bickhart et al. [43], because neither CNVs nor segmental duplications were observed in *BTF3*. Meanwhile, Angus cattle were treated as the reference sample for CNV identification. Moreover, CNV types were classified into three types that were gain type (Log_2_2^−ΔΔCt^ ≥ 0.5), loss type (Log_2_2^−ΔΔCt^ < −0.5) and medium type (−0.5 ≤ Log_2_2^−ΔΔCt^ < 0.5), thus, Log_2_2^−ΔΔCt^ was considered as CNV type value [26,30,31,41,42,43,44,45,46]. Finally, the CNV results were summarized and displayed by Prism software (version 5.0.0). In the analysis of gene expression profiles, internal reference Emerin gene (*EMD*) and lipoprotein receptor-related protein 10 gene (*LRP10*) were used as reference internal genes for the expression level identifications following the study of Liu et al. [47] and Saremi et al. [48], respectively. The gene expression levels were also standardized by the 2^−ΔΔCt^ method. Pearson correlation coefficient (PCC) of three replicates of tissue samples for gene expression levels was visualized by R package *corrplot* (version 0.84). The relative gene expression levels of different tissue samples were visualized in histograms by averaging three replicates.

### 2.5. Statistical Analysis

We used one-way analysis of variance (ANOVA) for all twelve phenotypes of NY, QC and XN cattle, separately, and two-way ANOVA for four common phenotypes of NY, QC and XN cattle together. Due to the commercial restrictions, cattle pedigree information was not available in six farms, so this study only applied the simple linear model rather than the linear mixed model considering the relationship matrix. The two models are:(1)$$Y_{i}=\mu +{\mathrm{CNV}}_{i}+e_{i},$$
(2)$$Y_{ij}=\mu +{\mathrm{Breed}}_{i}+{\mathrm{CNV}}_{j}+{\mathrm{Breed}}_{i}\times {\mathrm{CNV}}_{j}+e_{ij},$$
where Y is the phenotype, μ is the overall mean, Breed is the breed of cattle (i.e., NY, QC and XN breed), CNV is the CNV type (i.e., gain, loss and medium), and e represents residual errors. Pairwise multiple comparisons for testing the mean differences between different breeds, CNV types and their interactions were based on Tukey’s honestly significant difference (HSD) test [49].

### 2.6. Bioinformatics Comparisons of GBP6 Associated with Growth Traits

Based on the quantitative trait loci (QTLs) associated with growth traits of cattle along the whole *Bos taurus* genome (Bos_taurus_UMD_3.1), we investigated the density distribution of QTLs on different chromosomes. The QTL database was downloaded from Animal Quantitative Trait Loci Database (Animal QTLdb) for the growth traits of cattle (https://www.animalgenome.org/cgi-bin/QTLdb/BT/traitmap?trait_ID=1449&traitnm=Growth). Ten genes (*CYP4A11*, *GBP2*, *GPC1*, *KCNJ12*, *KLF3*, *KLF6*, *LEPR*, *MAPK10*, *MICALL2* and *MYH3*) identified in relationship with growth traits by previous CNV studies [23,24,25,26,27,28,29,30,31,32] were used to overlap with QTL database on different chromosomes and to investigate the relationship between growth trait-related QTLs and ten CNV-related genes. The QTLs with candidate genes were visualized by R package *RCircos* (version 1.2.0) [50]. Furthermore, the identified candidate genes from all studies were analyzed in DAVID (Database for Annotation, Visualization and Integrated Discovery) Bioinformatics Resources 6.8 (https://david.ncifcrf.gov/) to achieve significant pathways (*p* < 0.05).

## 3. Results

### 3.1. Animal Phenotypes

Due to the commercial limitations, not all the phenotypes for NY, QC and XN cattle were measured. Five phenotypes (BoH, BoL, BoW, ChC and HuW) for NY cattle, ten phenotypes (BoH, BoL, BoW, ChC, ChD, ChW, CrH, HuW, RuL and WaL) for QC cattle and seven phenotypes (BoH, BoL, BoW, CaC, ChC, CrH, and WaC) for XN cattle were collected in this study (Table 2). From the average values of four phenotypes (BoH, BoL, BoW and ChC) among NY, QC and XN cattle, we found that XN cattle had greater average body and chest size than NY and QC cattle, while NY and QC cattle showed similar performances (Table 2).

### 3.2. Distributions of the CNV in Six Breeds

We detected the copy number of *GBP6* in six breeds, in which 213 XN cattle, 112 NY cattle, 105 QC cattle, 32 DN cattle, 32 XJ cattle, and 30 ZS cattle were involved (Figure 3a). The CNV distributions were mainly close to zero, and the CNV distributions of XN cattle and ZS cattle were significantly different from NY cattle (adjusted *p* < 0.05) according to Student’s *t*-test after multiple testing correction using false discovery rate (FDR) (Figure 3a). Interestingly, XN cattle, ZS cattle and NY cattle are from three different regions where the average temperatures and precipitations are 14.9 °C and 979.2 mm, 9.2 °C and 527.9 mm and 14.9 °C and 777.7 mm, respectively (Figure 1). However, no significant difference of CNV distribution was observed among DN, NY, XJ cattle that are from neighboring regions with the same average temperature (14.9 °C) and precipitation (777.7 m) (Figure 1). After the identification of three CNV types, 174, 245 and 105 cattle displayed gain, medium and loss CNV types, respectively (Figure 3b). Most of NY and QC cattle exhibited the medium CNV type, whereas more gain CNV types than loss and medium CNV types were presented in XN cattle (Figure 3c). The frequency of relative CNV copy number increased and then decreased at the peak of two, as a result of most of relative CNV copy number was two (Figure 3d). Additionally, XN cattle has 40.65% (213/524) individuals of the whole analyzed samples, so it influenced the distribution of relative CNV copy numbers mostly, followed by NY cattle (112/524) and QC cattle (105/524) (Figure 3d).

### 3.3. Identified CNV Types of GBP6 Gene Associated with Growth Traits

Among NY, QC and XN cattle, the association analysis revealed that three phenotypes (i.e., BoW, CaC and ChC) of XN cattle performed significantly different across three different CNV types based on one-way ANOVA (*p* < 0.05) (Table 3). The results also showed that phenotypic values of BoW, CaC and ChC of XN cattle increased accordingly, when the CNV types were from loss to gain, especially for CaC trait with the significant phenotypic differences in different CNV types after pairwise comparisons (*p* < 0.01). Based on the two-way ANOVA, breed effects were significant for all four phenotypes (BoH, BoL, BoW and ChC) (*p* < 0.001), while CNV type and CNV type x breed interaction effects were not significant (*p* > 0.05) for these four phenotypes (Table 4).

### 3.4. Gene Expression Levels of GBP6

Three replicates of relative expression levels of *GBP6* showed high repeatability of the measurements with PCC values above 0.9 (*p* < 0.001) both in QC adults and XN adults (Figure 4a3,a4). However, the PCC values for three replicates of relative expression levels of *GBP6* were above 0.7 in QC fetuses (Figure 4a1) and lower than 0.5 in QC calves that showed poor repeatability of the measurements (Figure 4a2). Generally, tissue samples from spleen of QC and XN adults had higher relative expression levels than those from other tissues (Figure 4b). In the same stages, we used heart expressions as the control to compare with other tissue expressions. The results also showed that *GBP6* was significantly expressed in the muscle (*p* < 0.05) and spleen (*p* < 0.001) of QC fetuses, in the liver (*p* < 0.001) of QC calves, in the kidney (*p* < 0.01), liver (*p* < 0.01), muscle (*p* < 0.001) and spleen (*p* < 0.01) of QC adults, and in the liver (*p* < 0.05) and muscle (*p* < 0.001) of XN adults. In addition, *GBP6* was significantly expressed in the heart among three stages (*p* < 0.001), in the spleen, muscle and fat between fetal and adult stages (*p* < 0.05), and in the spleen between calf and adult stages (*p* < 0.05) of QC cattle after Student’s *t*-test (Figure 4b). Moreover, *GBP6* was significantly expressed in the heart (*p* < 0.05), kidney (*p* < 0.05) and lung (*p* < 0.01) of adult stage between QC and XN cattle (Figure 4b). The transcription levels of *GBP6* of *longissimus dorsi* muscle from XN cattle among three CNV types were strongly associated with growth traits (e.g., Cannon circumference (CaC) and body weight (BoW)) (Figure 4c). It was suggested that the gain CNV type of *GBP6* displaying the highest relative expression levels and resulted in highest phenotypes that included CaC (19.93 cm) and BoW (557.56 kg) (Figure 4c and Table 3). All of the relative expression values of *GBP6* in seven tissues of QC and XN cattle in fetal, calf and adult stages with three replicates were listed in Appendix A.

### 3.5. Comparative Validation of GBP6 with other Studies Associated with Growth Traits

The QTLs associated with growth traits of cattle were distributed on each chromosome and mainly concentrated on chromosome 6, 18 and 21. *GBP6* was located on chromosome 3 with low QTL densities, but it probably collaborates with the neighboring *GBP2* on affecting growth traits of cattle through potential strong linkage disequilibrium (LD) (Figure 5). Ten genes (*CYP4A11*, *GBP2*, *GPC1*, *KCNJ12*, *KLF3*, *KLF6*, *LEPR*, *MAPK10*, *MICALL2* and *MYH3*) on five different chromosomes (i.e., 3, 6, 13, 19 and 25) from previous similar CNV studies [23,24,25,26,27,28,29,30,31,32] were used for the comparison of our results (Table 5). These candidate genes were mainly located on chromosome 3 and enriched in the *Bos taurus* 04920: Adipocytokine signaling pathway (*p* < 0.05). They were consistent with body size traits (e.g., height, length and weight) of Chinese cattle from different provinces, where three of them exhibited positive associations between expressions and CNV types (i.e., gain CNV type with highest expressions), but six of them exhibited negative associations (Table 5).

## 4. Discussion

### 4.1. Domestic Beef Cattle in China

The six cattle breeds (NY, QC, XN, DN, XJ and ZS) in this study were the important breeds for beef production in China, especially for XN cattle which is a relatively new Chinese breed since 2007, after introducing Charolais cattle as sire to crossbreed with NY cattle for more than two decades [51]. Based on the hybridization for inheriting the good meat production from Charolais cattle [52], XN cattle exhibited higher body size values than NY cattle in this study (Table 2). Moreover, breed effect results in the two-way ANOVA showed that BoH, Bol, BoW and ChC traits of XN cattle were significantly higher than NY and QC cattle (*p* < 0.001) (Table 4). Mostly, the current genome-wide association studies (GWAS) apply pedigree information to control the population structure [53] and to estimate polygenic effects for the increased statistical power [54]. It is suggested the mixed model for GWAS fits pedigree in the kinship coefficient matrix to capture the polygenic variance [55], while SNPs or CNVs are tested as the fixed effects. Unfortunately, due to the restrictions in the commercial farms, no available pedigree data could be applied in this study; hence, the random variance accounting for the polygenic effects cannot be estimated.

### 4.2. The CNV Distributions with Biological Features of GBP6

The CNV distributions of *GBP6* revealed that the frequencies of three CNV types varied among six cattle breeds, especially for XN cattle owning more gain CNV type than the other cattle breeds (Figure 3). The results are consistent with the study of Shi et al. [31] that XN cattle also presented more gain copy number of *LEPR*. In one cattle species, CNV distribution could appear as an independent pattern, so the CNV copy number frequencies are different across diverse breeds [46]. As the crossbreed of Charolais and NY cattle, XN cattle tended to arise independent CNV patterns after a period of domestication for the breed formation. *GBP6* is an immune-related gene, which is located on chromosome 3 along with other four family members that are *GBP1* (54306891–54323003), *GBP2* (54345493–54437452), *GBP4* (54627197–54683058) and *GBP5* (54252901–54269516). The neighboring GBP genes are found to be located in the growth trait-related QTL regions (Figure 5), in which *GBP2* and *GBP4* within particular CNVRs that were associated with growth traits of Chinese cattle [24,36]. Interestingly, *LEPR*, *CYP4A11* and *GPC1* are also located on chromosome 3, so CNVs of these genes on the same chromosome are probably still in LD with a QTL because LD could exist at long distances [56].

### 4.3. The Identified CNVs and Expression Levels of GBP6 Associated with Growth Traits

The gain type of CNVs was reported in relationship with the increased body weight and chest size [27,31]. Our results also found that gain CNV type and highest body size traits (e.g., BoW, CaC and ChC) of XN cattle were in consistency (Table 3). Overall, SVs like CNV can cause differences in gene expression levels, and then influence the phenotype subsequently [2]. For example, some genes affected by the CNVs can change the copy-number to overlap or to disrupt the structure, and subsequently alter gene dosage in the genome [57]. Among one single gene, the insertions or deletions of CNVs directly influence the frameshifts, non-frameshift mutations and splice variants in the coding exon region. Additionally, the intron sequences can be affected by CNVs to result in alternative splicing [58].

Most importantly, the CNVs in cattle were also suggested to influence the genes for specific biological functions (e.g., immunity, rumination and reproduction) [46]. In addition, Stranger et al. [59] demonstrated that 17.7% of the total detected genetic variations in gene expression could be explained by CNVs, and the positive associations between CNVs and gene expressions were also observed in several studies [23,29,31,59,60]. In this study, the gain type of CNV might contribute *GBP6* expression levels to the highest in muscle tissues (Figure 4c); therefore, high expressions boosted the advantageous growth traits such as CaC and BoW. Transcriptome levels of *GBP6* varied significantly (*p* < 0.001) in different developmental stages and different tissues in the current study (Figure 4b), which indicated *GBP6* plays different important roles in the whole growth developments of cattle. By the combination of GWAS and global gene expression, the expression quantitative trait loci (eQTLs) study can identify QTLs that explain fractions of variation in the expression levels [61]. For example, QTLs associated with growth traits (Figure 5) can be used for eQTLs analysis by combining expression level of longissimus dorsi muscle with QTLs. In order to clearly define the presence of CNVs at genomic level influencing genome-wide gene expression levels related to the phenotypes on an individual, eQTLs analysis using CNVs could provide better understanding of basic genetic mechanism of gene expressions underlying growth traits [62].

## 5. Conclusions

Our study identified the CNV distributions of *GBP6* among six beef cattle breeds in China, and associated CNV types and gene expression levels with growth traits in XN, QC and NY cattle. Results showed that the CNV distributions varies among different cattle breeds and expression levels varies across different stages and different tissues of different breeds. Statistical analysis revealed that the CNV types were consistent with expression levels of *GBP6* and subsequently influenced the growth traits. It was suggested that *GBP6* could be applied in the breeding program for growth traits by including gain types of CNV.

## Figures and Tables

**Figure 1 animals-10-00566-f001:**
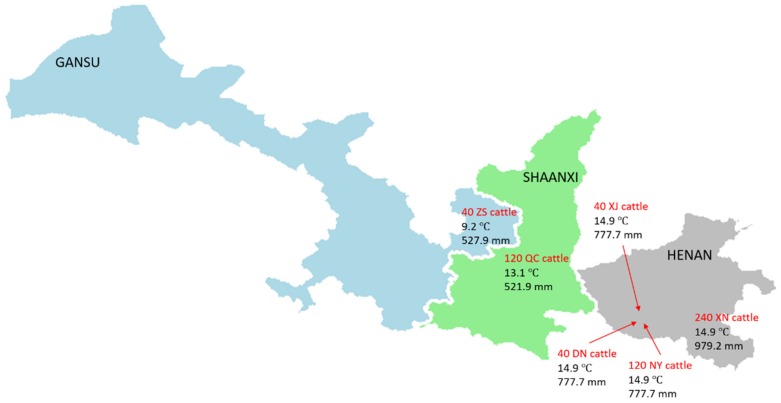
Map of regions for sample collections and their average temperatures (°C) and precipitations (mm) over past thirty years.

**Figure 2 animals-10-00566-f002:**
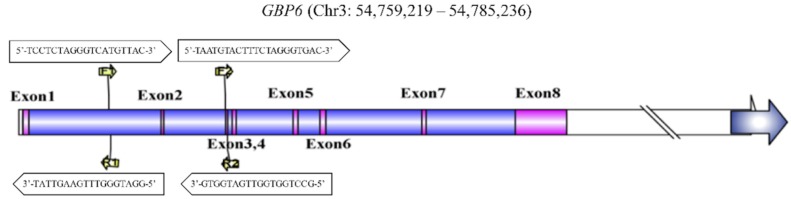
Two primers designed for quantitative polymerase chain reaction (qPCR) amplification on the gene structure of *GBP6*.

**Figure 3 animals-10-00566-f003:**
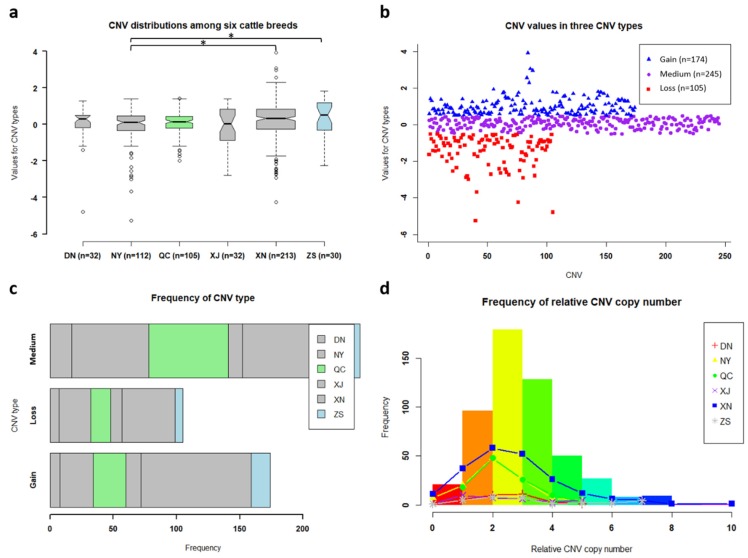
Copy number variations (CNVs) of *GBP6*. Note: SE: standard error; DN: Denan breed; NY: Nanyang breed; QC: Qinchuan breed; XJ: Xiajia breed; XN: Xianan breed; ZS: Zaosheng breed. (**a**) The distributions and statistics of CNV type values of *GBP6* among six cattle breeds. Note: * indicates adjusted *P* value < 0.05 according to Student’s *t*-test after multiple testing correction using false discovery rate (FDR). (**b**) Scatter plots of CNV type values in three CNV types (i.e., gain, medium and loss). (**c**) Frequencies of animals in three CNV types (i.e., gain, medium and loss) among six cattle breeds. (**d**) Frequencies of relative CNV copy number among six cattle breeds.

**Figure 4 animals-10-00566-f004:**
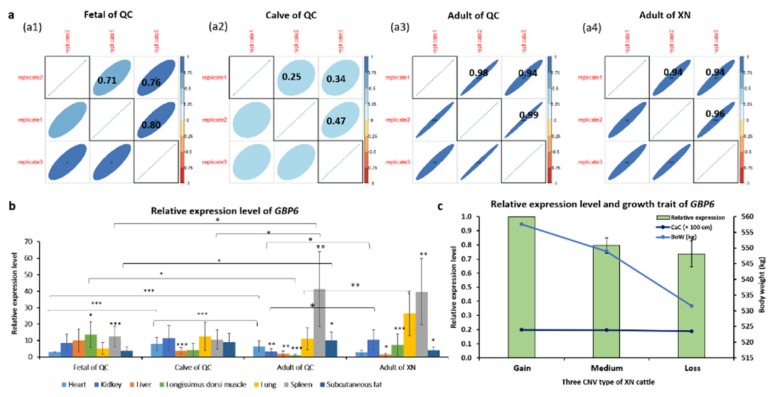
Relative gene expression levels of *GBP6*. Note: *, ** and *** indicates *p* value < 0.05, < 0.01 and < 0.001, respectively. QC: Qinchuan breed; XN: Xianan breed; CaC: Cannon circumference; BoW: body weight. (**a**) PCCs of three replicates of relative expression levels of QC fetuses (a1), QC calves (a2), QC adults (a3) and XN adults (a4). (**b**) Relative expression levels among seven tissues. (**c**) Relative expression levels of muscle tissues among three CNV types (i.e., gain, medium and loss) associated with growth traits of XN cattle. Note: Only BoW uses *y*-axis.

**Figure 5 animals-10-00566-f005:**
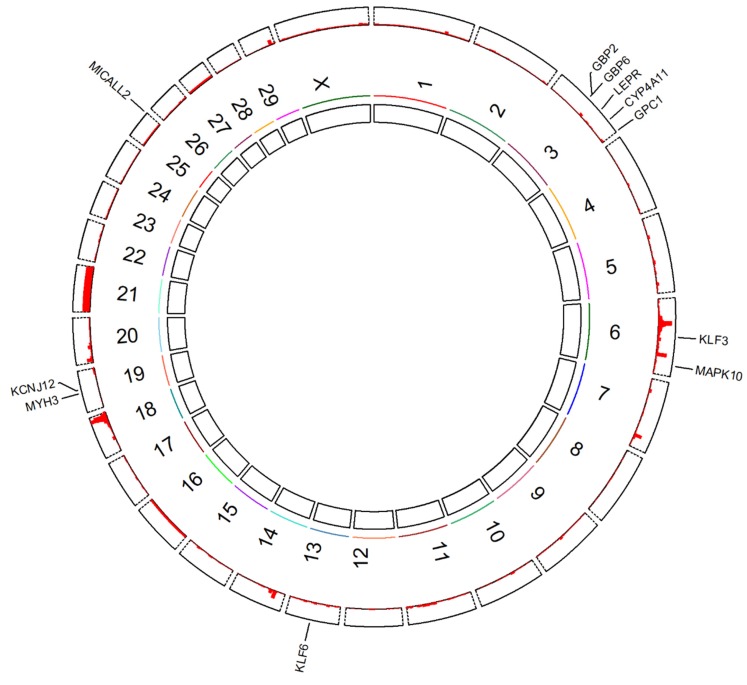
Circular graph of the densities of quantitative trait locis (QTLs) and identified candidate genes from other similar studies [23,24,25,26,27,28,29,30,31,32]. Note: the QTL densities in the red histogram were counted by 1 mega base pairs (Mb) windows.

**Table 1 animals-10-00566-t001:** Designed primers for real-time quantitative polymerase chain reaction (RT-qPCR) and qPCR based on *Bos taurus* genome of Bos_taurus_UMD_3.1 version.

Gene	Pairs of Primer Sequence (5’–3’)	Length (bp)
*GBP6* (1)	F-TCCTCTAGGGTCATGTTAC, R-GGATGGGTTTGAAGTTAT	90
*GBP6* (2)	F-TAATGTACTTTCTAGGGTGAC, R-GCCTGGTGGTTGATGGTG	110
*BTF3*	F-AACCAGGAGAAACTCGCCAA, R-TTCGGTGAAATGCCCTCTCG	166
*LRP10*	F-CCAGAGGATGAGGACGATGT, R-ATAGGGTTGCTGTCCCTGTG	139
*EMD*	F- GCCCTCAGCTTCACTCTCAGA, R- GAGGCGTTCCCGATCCTT	100

**Table 2 animals-10-00566-t002:** Statistics of Nanyang (NY), Qinchuan (QC) and Xianan (XN) cattle for twelve phenotypes.

Phenotype	Mean ± SE (Cattle Number)		
	NY (112)	QC (105)	XN (213)
BoH (cm)	126.81 ± 0.60 (58)	129.42 ± 0.58 (95)	135.52 ± 0.33 (198)
BoL (cm)	136.90 ± 0.89 (58)	137.39 ± 0.77 (95)	158.93 ± 0.51 (198)
BoW (kg)	373.10 ± 5.52 (58)	395.31± 6.17 (95)	548.88 ± 3.93 (198)
CaC (cm)	0	0	19.62 ± 0.13 (198)
ChC (cm)	169.67 ± 1.21 (58)	175.70 ± 1.01 (95)	193.68 ± 0.64 (198)
ChD (cm)	0	62.03 ± 0.47 (95)	0
ChW(cm)	0	37.62 ± 0.46 (95)	0
CrH (cm)	0	126.76 ± 0.59 (95)	138.59 ± 0.28 (198)
HuW(cm)	25.86 ± 0.27 (58)	22.42 ± 0.43 (95)	0
RuL (cm)	0	43.93 ± 0.29 (95)	0
WaL(cm)	0	42.40 ± 0.35 (95)	0
WaC (cm)	0	0	215.42 ± 1.45 (167)

Note: SE: standard error; NY: Nanyang breed; QC: Qinchuan breed; XN: Xianan breed; BoH: body height; BoL: body length; BoW: body weight; CaC: cannon circumference; ChC: chest circumference; ChD: chest depth; ChW: chest width; CrH: cross height; HuW: hucklebone width; RuL: rump length; WaC: waist circumference; WaL: waist length.

**Table 3 animals-10-00566-t003:** Association results of one-way ANOVA for phenotypes of NY, QC and XN cattle.

Breed	Phenotype	CNV type (Mean ± SE)			*p*-Value
		Loss	Medium	Gain	
NY	BoH(cm)	126.59 ± 0.51	126.96 ± 0.61	126.58 ± 0.71	0.96
NY	BoL(cm)	137.95 ± 0.73	136.89 ±0.91	135.96 ± 1.03	0.79
NY	BoW(cm)	381.00 ± 2.77	376.71 ± 5.57	355.33 ± 6.88	0.25
NY	ChC(cm)	171.27 ± 0.98	169.87 ± 1.16	167.62 ± 1.58	0.64
NY	HuW(cm)	25.77 ± 0.30	26.06 ± 0.26	25.38 ±0.32	0.63
QC	BoH(cm)	129.12 ± 0.55	129.75 ± 0.62	128.83 ± 0.50	0.79
QC	BoL(cm)	139.19 ± 0.80	136.71 ± 0.78	137.78 ± 0.75	0.50
QC	BoW(kg)	405.50 ± 6.03	388.07 ± 5.78	405.84 ± 7.13	0.38
QC	ChC(cm)	176.88 ± 0.91	174.58 ± 0.97	177.61 ± 1.17	0.41
QC	ChD(cm)	61.75 ± 0.55	61.89 ± 0.43	62.57 ± 0.53	0.82
QC	ChW(cm)	38.19 ± 0.47	37.22 ± 0.47	38.17 ± 0.45	0.60
QC	CrH(cm)	126.69 ± 0.53	126.82 ± 0.63	126.67 ± 0.55	0.99
QC	HuW(cm)	22.69 ± 0.38	22.04 ± 0.44	23.15 ± 0.44	0.54
QC	RuL(cm)	44.12 ± 0.28	43.77 ± 0.33	44.17 ± 0.22	0.82
QC	WaL(cm)	43.12 ± 0.26	41.89 ± 0.35	43.13 ± 0.40	0.22
XN	BoH(cm)	134.29 ± 0.46	135.61 ± 0.54	136.06 ± 0.40	0.13
XN	BoL(cm)	157.39 ± 0.64	159.24 ± 0.88	159.42 ± 0.61	0.30
XN	BoW(kg)	531.59 ± 3.69 ^a^	548.97 ± 4.01 ^ab^	557.56 ± 3.86 ^b^	0.04
XN	CaC(cm)	18.89 ± 0.10 ^Aa^	19.70 ± 0.12 ^b^	19.93 ± 0.14 ^B^	0.01
XN	ChC(cm)	189.93 ± 0.65 ^a^	194.70 ± 0.58 ^b^	194.62 ± 0.66 ^b^	0.01
XN	CrH(cm)	137.88 ± 0.24	138.92 ± 0.34	138.63 ± 0.23	0.40
XN	WaC(cm)	212.10 ± 1.32	216.03 ± 1.28	216.79 ± 1.39	0.44

Note: SE: standard error; NY: Nanyang breed; QC: Qinchuan breed; XN: Xianan breed; BoH: body height; BoL: body length; BoW: body weight; CaC: cannon circumference; ChC: chest circumference; ChD: chest depth; ChW: chest width; CrH: cross height; HuW: hucklebone width; RuL: rump length; WaC: waist circumference; WaL: waist length. ^ab/AB^ refers to significant difference after multiple pairwise comparisons (*p* < 0.05/0.01).

**Table 4 animals-10-00566-t004:** *p*-Value results of two-way ANOVA for four phenotypes of NY, QC and XN cattle.

Fixed effect	BoH			BoL			BoW			ChC		
Breed	<0.001			< 0.001			<0.001			<0.001		
	^A^ NY	^AB^ QC	^B^ XN	^A^ NY	^A^ QC	^B^ XN	^a^ NY	^Ab^ QC	^B^ XN	^A^ NY	^AB^ QC	^B^ XN
CNV type	0.29			0.81			0.25			0.19		
Breed × CNV type	0.78			0.42			0.12			0.09		

Note: ^ab/AB^ refers to significant difference after multiple pairwise comparisons (*p* < 0.05/0.01). NY: Nanyang breed; QC: Qinchuan breed; XN: Xianan breed; BoH: body height; BoL: body length; BoW: body weight; ChC: chest circumference.

**Table 5 animals-10-00566-t005:** Identified candidate CNV-related genes associated with growth traits of cattle from other studies.

Genes	Chromosome	Position	Growth Trait					Breed	Province of China	Association of CNV Type and Expression	Reference
			Body Height	Body Length	Body Weight	Heart Girth	Chest Girth				
*CYP4A11*	3	99,806,653–99,820,785				✓		Qinchuan	Shannxi	Positive	[23]
*CYP4A11*	3	99,806,653–99,820,785		✓				Nanyang	Henan	Positive	[23]
*CYP4A11*	3	99,806,653–99,820,785			✓	✓		Jinnan	Jilin	Positive	[23]
*GBP2*	3	54,345,493–54,437,452	✓	✓		✓		Pinan	Henan		[24]
*GPC1*	3	120,383,537–120,476,583	✓	✓	✓	✓		Datong yak	Qinghai	Negative	[26]
*KCNJ12*	19	35,955,260–35,993,796		✓	✓			Guangfeng	Jiangxi	Negative	[27]
*KCNJ12*	19	35,955,260–35,993,796		✓	✓			Jiaxian Red	Henan	Negative	[27]
*KLF3*	6	59,587,076–59,614,488			✓	✓		Cattle		Positive	[29]
*KLF6*	13	44,945,030–44,952,166	✓	✓	✓		✓	Datong yak	Qinghai	Negative	[25]
*LEPR*	3	80,071,689–80,167,592	✓	✓	✓		✓	Nanyang	Henan	Positive	[31]
*MAPK10*	6	102,683,802–103,320,828		✓	✓	✓		Nanyang	Henan	Negative	[32]
*MICALL2*	25	42,015,483–42,035,251	✓		✓		✓	Nanyang	Henan	Negative	[29]
*MYH3*	19	30,230,160–30,251,091	✓	✓	✓			Nanyang	Henan	Negative	[30]

Note: Positive/Negative indicate the expressions increase/decrease from loss to gain CNV types. ✓ indicates the associated growth traits.

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
