# Peer review of "Copy Number Variations and Expression Levels of Guanylate-Binding Protein 6 Gene Associated with Growth Traits of Chinese Cattle"

_animals, 2020, doi:10.3390/ani10040566_

Round 1
Reviewer 1 Report
The manuscript still contains many confusing and unclear statements. It requires a great deal of editing.
Please address the following questions and concerns.
lines 192-199 and Table 2: Maybe I am missing something, but I do not understand why a Chi-square analysis was used rather than an Analysis of Variance. Chi-square is generally used to analyze categorical data rather than continuous data.
lines 236-237: Change to "...while CNV type and CNV type x breed interaction effects were not significant (P > 0.05) for these four phenotypes".
Table 4: The first column does not show P-values. It shows the fixed effects. Therefore, the column heading needs to be changed. If the numbers shown in the table are P-values, the title of the table needs to be changed to reflect this fact. Currently, it is unclear what the numbers in the table represent.
lines 255, 260, and 277: Change "fetals" to "fetuses".
lines 263, 271, and 434; also in Supplementary table 1: Change "calve" to "calf".
Figure 4: The type in this figure is not readable. The same problem exists in some of the other figures as well.
line 289: What is meant by "good relationships". This seems to be a rather subjective term.
lines 291-292 and Table 5: Were correlation coefficients calculated or do you mean "associations" rather than "correlations"?
line 311: What is meant by "increment of statistical power"? Are you referring to increased statistical power?
line 320: What is meant by "In the specific cattle"?
line 344: "correlations" or "associations"?
line 362: The meaning of "CNV types were in consistent with expression levels" is unclear. Do you mean "were consistent with" or "were inconsistent with"?
Reviewer 2 Report
The structure of the paper is much improved, particularly with the new tables, and my concerns have now been addressed. I have a few remaining, very minor, comments.
Line 77. 'of tenderness' is not required.
Line 79. ‘nearly no study is’ should be ‘few studies are’.
Line 82. 'provide the valuable' should be 'provide valuable'.
I’m also still not entirely convinced that ‘medium CNV’ is a particularly accessible phrase, although given a definition is available (line 156), this isn’t much of a concern.
